# Morphologic and Molecular Landscape of Pancreatic Cancer Variants as the Basis of New Therapeutic Strategies for Precision Oncology

**DOI:** 10.3390/ijms21228841

**Published:** 2020-11-22

**Authors:** Chiara Bazzichetto, Claudio Luchini, Fabiana Conciatori, Vanja Vaccaro, Ilaria Di Cello, Paola Mattiolo, Italia Falcone, Gianluigi Ferretti, Aldo Scarpa, Francesco Cognetti, Michele Milella

**Affiliations:** 1Medical Oncology 1, IRCCS Regina Elena National Cancer Institute, 00144 Rome, Italy; chiara.bazzichetto@ifo.gov.it (C.B.); vanja.vaccaro@ifo.gov.it (V.V.); italia.falcone@ifo.gov.it (I.F.); gianluigi.ferretti@ifo.gov.it (G.F.); francesco.cognetti@ifo.gov.it (F.C.); 2Department of Diagnostics and Public Health, Section of Pathology, University and Hospital Trust of Verona, 37134 Verona, Italy; claudio.luchini@univr.it (C.L.); i.dicello@yahoo.com (I.D.C.); paolamattiolo@gmail.com (P.M.); 3Department ARC-Net Research Centre, University and Hospital Trust of Verona, 37126 Verona, Italy; aldo.scarpa@univr.it; 4Division of Oncology, University of Verona, 37126 Verona, Italy; michele.milella@univr.it

**Keywords:** pancreatic cancer, variants, histology, genetic status, molecular alteration, precision medicine

## Abstract

To date, pancreatic cancer is still one of the most lethal cancers in the world, mainly due to the lack of early diagnosis and personalized treatment strategies. In this context, the possibility and the opportunity of identifying genetic and molecular biomarkers are crucial to improve the feasibility of precision medicine. In 2019, the World Health Organization classified pancreatic ductal adenocarcinoma cancer (the most common pancreatic tumor type) into eight variants, according to specific histomorphological features. They are: colloid carcinoma, medullary carcinoma, adenosquamous carcinoma, undifferentiated carcinoma, including also rhabdoid carcinoma, undifferentiated carcinoma with osteoclast-like giant cells, hepatoid carcinoma, and signet-ring/poorly cohesive cells carcinoma. Interestingly, despite the very low incidence of these variants, innovative high throughput genomic/transcriptomic techniques allowed the investigation of both somatic and germline mutations in each specific variant, paving the way for their possible classification according also to specific alterations, along with the canonical mutations of pancreatic cancer (*KRAS*, *TP53*, *CDKN2A*, *SMAD4*). In this review, we aim to report the current evidence about genetic/molecular profiles of pancreatic cancer variants, highlighting their role in therapeutic and clinical impact.

## 1. Introduction

Among all the exocrine malignant pancreatic cancers, pancreatic ductal adenocarcinoma (PDAC) occurs with the highest percentage of frequency (85%) and represents one of the most lethal tumors, with a 5 year-overall survival (OS) of 5–10% [1,2,3]. The dismal outcome of this cancer is mainly due to the lack of early diagnosis, which means that at the moment of the diagnosis, patients are not eligible for surgical resection and present advanced metastatic disease. Furthermore, as opposed to other solid malignancies, precision medicine does not represent a standard of care. Indeed, the cornerstone in the treatment of pancreatic cancer is still the aggressive and cytotoxic chemotherapy, as target therapies met with limited clinical success, mostly due to the absence of validated prognostic/predictive biomarker(s) [4]. The reason behind this failure is due, at least in part, to the complexity of the pancreatic tumor for both its genetic status and bidirectional interactions with the tumor microenvironment (TME) [5]. Indeed, pancreatic cancer cells are characterized by a hypermutated landscape, with four most commonly mutated genes: the *Kirsten rat sarcoma* (*KRAS*) oncogene, the *tumor suppressor protein 53* (*TP53*), the *cyclin dependent kinase inhibitor 2A* (*CDKN2A*), and the *SMAD family member 4* (*SMAD4*) tumor suppressor genes [6]. All these mutations are involved in the dysregulation of core signaling pathways, which affects not only specific features of tumor cells (i.e., proliferation and migration), but also the crosstalk with their surrounding desmoplastic TME [6]. Along these so-called “pancreatic genetic mountains”, genomic analysis also revealed a milieu of other genetic and chromosomal alterations, such as mutations in *AT-rich interaction domain 1A* (*ARID1A*), *breast cancer gene* (*BRCA*) *1*, and *BRCA2*. In 2016, a large study including 456 pancreatic cancer patients classified these tumors into four subtypes according to a comprehensive integrated genomic and transcriptomic analysis; very interestingly, these subtypes are associated with specific histopathological characteristics [7]. Histology is the feature by which the World Health Organization (WHO) currently classifies PDAC into eight variants: colloid carcinoma (CC), signet-ring cell carcinoma (SRCC), undifferentiated carcinoma with osteoclast-like giant cells (UCOGC), adenosquamous carcinoma (ASC), undifferentiated carcinoma (UDC), and hepatoid, medullary, and rhabdoid carcinoma (Figure 1) [8].

As highlighted above, very recent published papers characterized these variants also according to their genetical status and subsequent molecular pathway’s activation. As shown in Table 1, indeed, all the pancreatic cancer variants are characterized by alterations in conventional driver genes, very low incidence, and quite poor prognosis.

Hence, a more comprehensive analysis of these variants would be very helpful in the clinical setting, as they help to distinguish between apparently similar tumors, which could display completely different outcomes. Consistently, subtyping pancreatic cancer with comprehensive and specific genetic-molecular-morphological characteristics could help clinicians to define the best therapeutic choices and improve patient clinical outcome [42].

In this review, we want to give an update of the current knowledge of the genetic and molecular landscape of each PDAC variant, underlining that this evidence should be improved in order to better design clinical trials and define new PDAC targeted therapies.

## 2. Genetic and Molecular Features of PDAC Variants

According to the most recent WHO classification, PDAC variants can be categorized into eight variants with specific histomorphological features (Figure 1) [8]. Here, we summarize the current knowledge about each genetic background of PDAC precursors (Table 2).

### 2.1. Colloid Carcinoma

Also known as mucinous non-cystic carcinoma, CC is characterized by abundant stromal mucin pools with suspended neoplastic cells (50–80%) (Figure 1). Usually, neoplastic cells are structured in small aggregates inside mucin pools, but they may be also isolated. Histologically, mucin pools are often surrounded by a chronic inflammatory response rich in lymphocytes [43]. CC often occurs at the head of the pancreas and derives from the intestinal-type intraductal papillary mucinous neoplasm (IPMN) precursor lesions [44,45]. In particular, Yamada et al. showed that CC displays *GNAS* mutations, similarly to intestinal villous adenomas [46]. Subsequently, Amato and her colleagues investigated the mutational status of 52 intraductal neoplasms of the pancreas, by next generation sequencing (NGS) technique. Among the IPMN intestinal type, 83% harbor mutations in *GNAS* [47]. Consistent with the histological CC characteristics, in vitro experiments showed that *GNAS* mutations alter gene expression profiles, particularly the expression of mucin genes, like *MUC2* and *MUC5AC* [48]. Nevertheless, despite pancreatic CC displays intracytoplasmatic MUC2, positive MUC2 staining does not specifically characterize pancreatic CC [49].

Very frequently, the *GNAS* gene in chromosome 20 displays somatic mutations in CC, mostly in the codon 201, resulting in R201H or R201C variants in the catalytic domain of the protein [50]. These missense mutations gain oncogenic capabilities to the protein, which is constitutionally active. Indeed, the encoded protein is the Gα subunit of heterotrimeric G-proteins, responsible for integrating extracellular stimuli from the G-proteins-coupled receptors (GPCR) within the cell [51]. Once activated, the Gα subunit catalyzes the exchange of guanosine diphosphate (GDP) in guanosine triphosphate (GTP), thereby resulting in the dissociation from the β and γ subunits and effectors engagement. The main actors in GPCR-mediated signaling are the adenylyl cyclase and its direct product, the second messenger cyclic adenosine monophosphate (cAMP), which in turn activates the protein kinase A (PKA). The transcription factor cAMP-response element binding protein (CREB) represents the central hub of the PKA signaling; once phosphorylated, p-CREB interacts with other transcriptional co-activators, thereby enhancing the transcription of several genes involved in conferring tumorigenic properties to the cells (i.e., migration and invasion abilities) [52,53]. Other PKA-downstream elements are the transcriptional activator yes-associated protein (YAP), and the tumor promoters extracellular signal-regulated kinase (ERK) 1/2 and β-catenin. Indeed, PKA phosphorylates and hence stabilizes β-catenin, which translocates into the nucleus to enhance the transcription of oncogenes, like *c-myc* and *cyclin D1* [54,55]. Furthermore, also non-canonical PKA signaling is activated downstream cAMP. Indeed, the two isoforms of the exchange protein directly activated by cAMP (EPAC) 1 and 2 contain the cAMP-binding domains, by which EPAC1/2 regulate cAMP-related functions, such as cell adhesion differentiation and proliferation [56].

The central role of *GNAS* mutations in CC was confirmed by other studies, that also provided significant data that *GNAS* and *KRAS* mutations are two different markers in CC and tubular carcinoma, respectively [57,58]. Nevertheless, literature data also reported that driver mutations in both *GNAS* and *KRAS* co-occur in a subset of pancreatic cancer patients. In particular, taking the advantage from genetically engineered mice models, Patra and collaborators demonstrated some important evidence in the genetic and metabolic behavior of malignant IPMN [59,60]. First, the authors showed that *GNAS^R201C^* synergizes with *KRAS^G12D^* in establishing tumorigenesis processes of IPMN, which rapidly evolve in PDAC once *TP53* is inactivated. Nevertheless, amid the mutations in these three genes, *GNAS^R201C^* displays a predominant role in tumor sustaining, via cAMP-PKA signaling. In particular, GNAS-PKA activation phosphorylates and inhibits salt-inducible kinases (SIK) 1 and SIK3, a family of AMP-activated protein kinase (AMPK)-related kinases. The suppression of SIK1 and SIK3 results in the induction of lipid remodeling and fatty acid oxidation. Furthermore, the treatment with forskolin, an adenylyl cyclase agonist, increases cAMP levels rescuing *GNAS*-silenced organoid growth [60].

In CC, mutations in the *ataxia telangiectaisa mutated* (*ATM*) gene often occur, as reported by Hutchings et al. Indeed, the authors demonstrated that 13% of patients with germline *ATM* pathogenic variants display CC. Although the number of patients were very limited, this was the first study which correlated *ATM* germline status and CC [61]. By the genetic analysis of a series of familial pancreatic cancer probands, Roberts and his group identified deleterious *ATM* variants as putative pancreatic cancer susceptibility genes [62]. *ATM*, localized in chromosome 11, encodes for a serine/threonine kinase involved in the repair of the deadly DNA double strand break [63]. The Mre11/Rad50/NBS1 (MRN) complex recruits ATM in the region where the double strand break occurs and leads to the activation of the ATM-mediated phosphorylation cascade, to induce cell cycle arrest, apoptosis, or senescence [63,64]. Due to this central role in cell death, *ATM* is often mutated and inactivated during neoplastic transformation. Consistently, patients affected by ataxia telangiectasia, a severe syndrome due to biallelic mutations in *ATM*, display increased susceptibility to different types of cancer, including hereditary pancreatic cancer [65]. Interestingly, somatic *ATM* deficiency increases DNA damage in *KRAS*-mut precancerous lesions, transformed cells acquire new oncogenic mutations and pancreatic tumor is overt. This evidence establishes a predominant role for *ATM* in maintaining the genomic stability of pancreatic cells [66]. For all this molecular evidence, *ATM* status analysis displays clinically relevant therapeutic implications. Indeed, similar to *BRCA* genetic alterations, *ATM* aberrations increase the efficacy of chemotherapeutic drugs which induce double strand DNA breaks, such as platinum drugs [67,68].

Another key aspect in genomic instability is represented by the mismatch repair (MMR) deficiency-dependent microsatellite instability (MSI). The MMR genes, *MLH1*, *MLH2*, *PMS2* and *MSH6*, often harbor germline mutations, although they occur at low levels in pancreatic cancer (2%). In that respect, Lupinacci and his group recently investigated the MMR status in 445 PDAC specimens, by analyzing the mutational landscapes of these genes. Despite the fact that the authors detected MMR deficiency in only 1.6% of samples, the loss of MMR genes was higher in IPMN tumors as compared to non-IPMN ones (*p* = 0.02) [69]. MSI leads to the expression of immunogenic neo-antigens by cancer cells, hence resulting in an immunogenic phenotype with high number of infiltrating CD8^+^ T-cells and high expression of immune checkpoint molecules (e.g., programmed cell death protein (PD)-1 and PD-ligand (PD-L) 1). Interestingly, MMR-deficiency appears to display a prognostic value in the treatment of these types of cancer. In particular, this MSI-dependent immunogenicity promotes the sensitivity to immune checkpoint inhibitors in solid tumors, including pancreatic cancer [70]. Due to the increased interest in MSI and immunotherapy, a recent meta-analysis aimed to uniquely correlate MSI and histological features of PDAC patients. By analyzing 8323 patients, the paper covers two important points: (I) the relevance of validated methods for MSI assessment; and (II) the real concreteness of correlating genetic/molecular characteristics with histopathological ones. In particular, analysis of data derived by only standardized and validated NGS techniques shows that MSI/MMR deficiency occurs with an even lower prevalence, around 1%. Intriguingly, this molecular alteration is strongly related to the colloid histology (*p* < 0.01) [16].

### 2.2. Medullary Carcinoma

Despite a very low number of papers that have discussed the molecular characteristics of medullary carcinoma, it is recognized that in this poorly differentiated variant, MSI also occurs with a high percentage [16,71]. Consistently, the medullary phenotype of pancreatic cancer is characterized by the presence of syncytial growth pattern, necrosis, and expanding tumor borders (Figure 1) [43,72,73]. Interestingly, a gland component is usually totally absent, but in rare cases it may be present [74]. In a seminal paper in 1998, Goggins and coworkers described the medullary carcinoma as a distinct subset of pancreatic adenocarcinoma for the first time. The authors concluded that the medullary phenotype could specifically identify germline MMR mutated patients, who can benefit from genetic counseling for the identification of carriers of Lynch syndrome. Moreover, as these patients present DNA replication errors, this variant might be associated with a better prognosis, despite the apparently poor differentiation of the tumor [72]. In order to better define this subset of patients, the same group further investigated the genetic characteristics of medullary carcinoma derived from both patients and xenografts [14]. Interestingly, results revealed a higher percentage of *KRAS*-wt status (67%), as opposite to canonical *KRAS*-mut pancreatic cancer, as also confirmed by a comprehensive systematic review coupled with a comparative analysis with large datasets [16]. MSI percentage in this PDAC variant was around 20%. Among MSI-positive tumors analyzed by immunohistochemistry (IHC), the authors identified a specific loss of MLH1, and not MSH2, expression; on the contrary, non MSI-tumors express both MLH1 and MSH2. This evidence clearly suggests that MLH1 and MSH2 IHC could be useful biomarkers to identify medullary carcinoma patients with MSI [14]). As MMR deficiency mainly results from *MLH1* hypermethylation, Kondo and collaborators investigated MLH1 expression levels in pancreatic and endometrial cancer, both characterized by high MSI. Nevertheless, these results show that *MLH1* is not silenced in pancreatic cancer, as opposite to endometrial tumors, hence highlighting that other, non MLH1-dependent, mechanisms are the basis of DNA instability in medullary carcinomas [75]. Consistently, for the first time in 2006, Banville reported a case of medullary carcinoma occurring in a patient with hereditary nonpolyposis colorectal cancer (HNPCC, Lynch syndrome), with germline mutations of *MSH2*. This evidence once more suggests that many efforts still need to be made to better characterize the genetic alterations underlying MSI in medullary pancreatic tumor [76]. A recent comprehensive review clarified that all four classic MMR genes can be involved in the pathogenesis of PDAC-medullary variant [16].

A recent case report details the first case of a woman affected by medullary pancreatic cancer, with a good prognosis. This was characterized by a very high tumor mutational burden but did not show MSI; notably, a somatic mutation in the *polymerase epsilon* (*POLE*), which can explain the very high tumor mutational burden, was detected [13]. The *POLE* gene, localized in chromosome 12, encodes for the catalytic subunit of the eucaryotic DNA polymerase, involved in DNA replication and repair [77]. *POLE* mutations often occur in its exonuclease proofreading domain, hence resulting in DNA damage, similar to an MMR status [78]. Consistently, the authors conclude that *POLE* mutations could represent an alternative molecular pathway in medullary carcinoma, with promising implications in treatment and prognosis [13].

Along genetic features, the medullary carcinoma is one of the variants, in addition to UCOGC, which can be characterized also by specific TME infiltration. Indeed, a recent case report described an abundant lymphocytes infiltration between tumor and non-tumor areas [15]. Moreover, an interesting work also describes a marked lymphocyte infiltration in medullary pancreatic adenocarcinoma samples, through IHC analysis [16].

### 2.3. Adenosquamous Carcinoma

Opposite to the good prognosis of the medullary carcinoma, pancreatic ASC displays aggressive metastatic potential and thus the worst patient prognosis [79]. First described by Madura and colleagues as “cancroide”, this variant affects the exocrine area of the pancreas and is characterized by both adenomatous and keratinized squamous cells (Figure 1) [80]. By definition, to be characterized as “adenosquamous”, the squamous component must be at least 30% of the entire lesion. Histologically, peri-neural invasion and lympho-vascular invasion are very common, and cell atypia is usually very high, above all in the squamous component [43]. Owing to a low frequency of 1–4% of all pancreatic neoplasms, very little is known about the driver alterations in the pathogenesis of the disease [81]. Results obtained from whole genome analysis highlight mutations mainly in conventional genes, such as *MYC*, *SMAD4*, *phosphatase and tensin homolog* (*PTEN*), and *TP53* [18,19,82]. Hence, the main feature of this tumor is the lack of specified related gene mutations, as alterations in *KRAS* and *TP53* are also typical of other variants of pancreatic cancer [17,20]. Indeed, no compelling evidence on genetic and molecular profiling of ASC is reported.

In 2014, Liu and coworkers published the first data about somatic mutations in *up-frameshift* (*UPF*) *1* in this pancreatic tumor lesion [83]. *UPF1*, localized in chromosome 19, encodes a cytoplasmatic RNA helicase involved in nonsense-mediated mRNA decay (NMD). NMD is a complex surveillance mechanism to eliminate mRNAs containing premature stop codons [84]. In ASC, *UPF1* point mutations, detected in helicase and carboxy-terminal region, disrupt *UPF1* splicing, hence promoting alternative *UFP1* precursor mRNA. NMD perturbation could be the root of the high malignant behavior of the ASC: for example, the protein encoded by an alternatively spliced TP53 transcript, the alt-PTC-IVS6-p53, harbors dominant-negative activity [83]. Unlike the data reported by Liu et al., Hayashi and coworkers didn’t observe any mutations in *UPF1* in their clinical samples. Nevertheless, the authors identified interesting genetic alterations in chromatin modifier genes. In particular, mutations in the *histone-lysine N-methyltransferase 2 (KMT2*)*C* and *KMT2D*, the ATP-dependent chromatin remodeler *SMARCA4*, and *lysine demethylase* (*KDM*)*6* are reported in 7%, 5%, and 5% of the patients, respectively [85]. A recent manuscript by Lenkiewicz and collaborators showed mutations also in the *KDM3* [86]. Alongside confirming the alterations in chromatin regulatory genes, this manuscript is very relevant for two critical aspects: (I) it confirms that ASC evolves from the same lineage as PDAC, as they share canonical mutations in *KRAS, TP53*, *CDKN2A*, *SMAD4* and *MYC;* and (II) it demonstrates that the genetic characterization of the PDAC variants is crucial in defining new therapeutic intervention strategies. By ATAC-seq in KPC mouse model, the authors define the status of the *nuclear receptor ROR-γ* (*RORC*)—a nuclear receptor hormone involved in the Th-17-dependent inflammatory cytokines release—as a specific feature of ASC; the authors also demonstrated the co-occurrence of genetic alterations in *KRAS* and *fibroblast growth factor receptor* (*FGFR*). From an interesting translational point of view, the authors showed that organoids *RORC*^+^/*FGFR-ERLIN2* fusion/*KRAS^G12V^* are more sensitive to the single-agent FGFR inhibitor infigratinib [86].

Despite the absence of genetic signature for ASC, two recent papers investigated the immunological mutational signature which could display interesting implications in terms of cancer treatment. Silvestris and his group showed that 15% of ASC express PD-L1, and, interestingly, PD-L1 positive tumors display squamous histology [87]. Another interesting paper demonstrates that pancreatic ASC displays higher PD-L1 levels, as compared to conventional PDAC. More in particular, the authors confirm that PD-L1 expression is detected only in the squamous component [88]. This evidence is particularly relevant for the possibility to stratify patients as PD-L1 negative or positive, hence as immune-checkpoint inhibitors non-responder or responder, respectively. An added marker of squamous differentiation is DeltaNp63 (DNp63), an isoform of the p63 protein (a member of the p53 family) which lacks the transactivation domain. DNp63 is specifically detected in stem cells-like phenotype, whereas canonic PDAC is often negative for the IHC expression of DNp63. This squamous specificity of DNp63 could explain the metastatic aggressiveness of this subtype of pancreatic cancer [89].

Another important feature characterizing potential biomarkers to improve the cancer biological therapeutic opportunities is the formation of new vessels. Despite PDAC being defined by a low microvascular density, no sufficient data are reported for angiogenesis in ASC. Interestingly, through the analysis of the expression of a specific set of genes and miRNAs, Silvestris and his coworkers demonstrated a greater number of micro vessels in ASC, as compared to conventional PDAC, thus suggesting another keystone in personalized treatment [90].

### 2.4. Undifferentiated Carcinoma

UDC, also called anaplastic carcinoma, is a rare pancreatic cancer variant (1–7% of frequency of ductal adenocarcinoma), principally localized in the head of the pancreas and often associated with worse prognosis. UDC is characterized by the lack of a defined differentiation as demonstrated by the presence of pleomorphic mononuclear cells interspersed with rhabdoid or spindle cells, and several morphological variants (i.e., anaplastic, sarcomatoid) (Figure 1) [24,91].

Regarding molecular characterization, the delineation of UDC features has been described since 1998. Indeed, Gansauge and collaborators described a significant upregulation of TP53 and CD95 in undifferentiated tumor through IHC [28]. Despite its canonical role in mediating apoptosis as a death receptor, the controversial role of CD95 is now well known in cancer cells. Indeed, CD95-mediated signals are involved in cell cycle progression even in PDAC [92]. Moreover, Winter and collaborators evaluated the level of E-cadherin in pancreatobiliary cancers with UDC. All of the analyzed UDC display the loss of E-cadherin (93% and 100% of anaplastic and osteoclast-like giant cells, respectively), probably due to the hypermethylation of *CDH1* (the gene codifying for E-cadherin), as shown in in vitro pancreatic UDC cell line [93]. E-cadherin belongs to the cadherins family, and is involved in the formation of adherens junctions and responsible for maintaining an epithelial phenotype. Conversely to E-cadherin, a positive correlation between UDC and vimentin and zinc finger E-box binding homeobox 1 (ZEB1) is observed by IHC [94,95]. Indeed, vimentin is a marker of mesenchymal features, while ZEB1 represents a transcriptional repressor of epithelial-to-mesenchymal transition (EMT) induction [96,97]. Since E-cadherin deficiency and vimentin/ZEB1 upregulation are associated with EMT mechanisms, it is not surprising that UDC is a particularly aggressive variant [98].

The dedifferentiation can occur either during cancer progression or as a result of the treatment’s pressure; specific molecular clusters, downregulating epithelial and squamous differentiation and upregulating KRAS, but not TP53, signaling are at the base of dedifferentiation [99]. Krasinskas and her group demonstrated that the progression from PDAC to UDC was associated with changes in *KRAS* copy number. Indeed, in the analyzed cohort, the 73% of *KRAS* mutant allele-specific imbalance (due to the loss of wild type (wt) *KRAS*-carrying chromosome or hyperploidy of the mutated ones) is associated to UDC [27].

Despite all the reported genetic/molecular features, UDC still lacks specific genetic signature and is hence often superimposable on PDAC, as they do not significantly differ from a biological point of view.

### 2.5. Undifferentiated Carcinoma with Osteoclast-Like Giant Cells Carcinoma

Among UDC, a new variant is arousing growing interest given the presence of specific characteristics both at a morphological and molecular level. Indeed, we distinguish the UCOGC for some specific features. First, they are indeed composed of three distinct types of cells: (I) mononuclear histiocytes, which are inflammatory cells (type-2 tissue associated macrophages, CD163-positive) that stimulate tumor growth and proliferation; (II) osteoclast-like giant cells, which are inflammatory cells with phagocytic activity and can contain up to 30 nuclei; and (III) mononuclear neoplastic cells, which are highly atypical and pleomorphic elements that show enlarged nuclei with prominent nucleoli (Figure 1) [43,100]. UCOGC represents a PDAC variant with a very low incidence (<1%), which seems to arise from ductal lesions [101,102,103]. Even if the presence of osteoclast-like giant cells is well documented in tissues with different histological origin (e.g., skin, kidney, breast), the pancreas is the most common site for UCOGC. Notably, we had to wait until 2010 for the recognition of UCOGC as a PDAC variant for both epithelial origin and pathological characteristics, by WHO [104]. Hence, many of the UDC studies published before 2010 considered UDC and UCOGC as the same variant (such as in [93], previously described). For example, Bergmann and his group analyzed the expression of an anaplastic carcinoma cohort, half of which contained osteoclast like-giant cells, to evaluate the presence of potential druggable targets. Indeed, the study highlights a significant presence of canonical targets of well-known therapies, such as L1CAM, cyclooxygenase (COX) 2, and epidermal growth factor receptor (EGFR) in 80%, 93%, and 87%, respectively [105]. In a study which comprises 38 resected UCOGC and 725 resected PDAC, Muraki and collaborators demonstrated that the UCOGC is characterized with larger tumor size and is clinically diagnosable mainly in younger patients as compared to the other PDAC. Moreover, it is also associated with a better prognosis and a significant increased survival [33]. The importance of histology was also confirmed in a recent study that showed a better prognosis in case of “pure” UCOGC (i.e., not associated with a PDAC component) compared with UCOCG associated with a glandular component [31].

Amid the most mutated genes of this variant, it is necessary to mention the conventional *KRAS*. Indeed, several studies described mutations of this oncogene in different UCOGC cohorts or case reports [30,106,107]. Moreover, in order to better identify specific alterations of this rare variant, additional genetic features are currently under investigation. A recent analysis of a cohort of patients with UCOGC, characterized through whole exome sequencing (WES), showed non-synonymous somatic missense mutations in *serpin peptidase inhibitor clade A member* (*SERPINA*) *3*, *melanoma-associated antigen* (*MAGE*) *B4*, *glioma-associated oncogene* (*GLI*) *3*, *multiple epidermal growth factor-like domains protein* (*MEGF*) *8* (each mutated in two patients), and *TTN* (mutated in three tumors). More in particular, two patients share the same missense mutation in *SERPINA3* in a hotspot region which could indicate its possible role as oncogene [31]. *SERPINA3* encodes for a homonymous protein, also known as α−1-antichymotrypsin, which belongs to the serine peptidase inhibitors family. Interestingly, SERPINA3 is a useful biomarker in the prognosis of different solid tumors, including endometrial cancer: in this histotype, SERPINA3 expression is significantly correlated with the worst pathological grade, lymph node metastasis (LNM) grade, and clinical stage [108]. Similar poor prognosis is also observed in melanoma [109]. It is also known that IHC staining for SERPINA3 is higher in colon cancer tissue compared to normal tissue; moreover, *SERPINA3* silencing by siRNA reduces the level of migration, invasion, metastases, and the expression of metalloproteinase in in vitro and mice models of colorectal cancer (CRC) [110].

Unlike *SERPINA3* mutations, the other reported mutations (i.e., *MAGEB4*, *GLI3*, *MEGF8*, and *TTN*) are all non-synonymous missense mutations not in a specific hotspot region [31]. *MAGE* genes are mainly active in the processes of embryogenesis and are subsequently switched off by epigenetic processes, such as hypermethylation. Their reactivation during neoplastic transformation may play an important role in immunosurveillance, as MAGE proteins are antigens expressed by several malignancies [111]. Although their expression is not yet well investigated in PDAC, a recent study showed that high expression levels of MAGEB4 mRNA in breast cancer patients correlate with an improved relapse-free survival, regardless of breast cancer subtype [112]. *GLI3* and *MEGF8* belong to the well-known hedgehog (HH) signaling cascade, frequently altered in PDAC [113]. The canonical HH cascade is composed of three different ligands, sonic hedgehog, Indian hedgehog and desert hedgehog, which bind two possible receptors protein patched homolog (Ptch) 1 and Ptch2. After the ligand-receptor binding, the signal transducer smoothened (Smo) activates the GLI transcription factors [114]. Indeed, GLI3 represents one of the main targets and transductor of HH pathway. In absence of ligand and through the proteolysis mechanism, GLI3 acts as a HH repressor [115]. A recent work by Ma and collaborators demonstrated that GLI3 is significantly correlated with distant metastasis, vascular invasion, and histologic grade in pancreatic cancer patients [116].

As opposite to GLI3, the transmembrane protein MEGF8, which interacts with mahogunin ring finger-1 (MGRN1) for the ubiquitination and degradation of Smo, is not correlated with pancreatic cancer [117]. Likewise, limited evidence is reported for *TTN*, although it is now recognized as a tumor associated gene in several cancers including pancreatic cancer [118]. Indeed, Wu and collaborators recently demonstrated that the level of TTN mRNA, together with seven other mRNA targets and eight lncRNAs, is significantly associated with a PDAC patient’s OS [119]. Overall, due to the presence of *SERPINA3* and *GLI3* alterations, evaluating the mutational status of these genes as specific markers of UCOGC could be a novel strategy in better characterization of this variant.

Meanwhile, due to the current lack of specific markers for this variant, many other case reports investigating the genetic pattern of UCOGC patients are currently evaluated. For instance, a recent work by Yang and coworkers reports a case which displayed *BRCA2* somatic mutation, in addition to the *KRAS^G12D^* somatic mutation, by WES [36]. *BRCA 1/2* are tumor suppressor genes, involved in DNA maintenance and frequently mutated in familial pancreatic cancer. Indeed, BRCA 1/2 are involved in homologous recombination repairs of double strand breaks together with other proteins such as PALB2, ATM, and RAD50 [120]. Therefore, it is not surprising that up to 17.4% of PDAC is characterized by “BRCAness” signature genes which imply the use of olaparib in presence of *BRCA* mutations [121].

Nevertheless, despite all the described alterations in this specific variant, the current knowledge on genetic characterization of UCOGC still remains insufficient to uniquely correlate UCOGC and specific driver genes. Hence, interesting papers aimed to characterize UCOGC from a molecular point of view. Although Slug is the most important marker in UDC and UCOGC variants, UCOGC differs from UDC for the levels of EMT markers (i.e., Twist1, Slug, and E-cadherin) expression. Indeed, a recent paper demonstrated that UDC displays a significantly higher activation of EMT as compared to UCOGC (100% vs. 50% of cases, respectively). However, the levels of EMT in UCOGC increase if UCOGC is associated with a PDAC component (i.e., conventional genetic alterations) [122]. As the immune system and the immune inhibitory pathways display relevant clinical implications, a compelling paper finely described the state of PD-L1 and PD-1 in a series of 27 UCOGC patients (of which 16 were PDAC-associated). PD-L1 expression was present in 63% of UCOGC cases; interestingly, by considering only PDAC-associated UCOGC, the expression rise up to 81% of cases. Furthermore, while the expression of PD-L1 in peripheral lymphocytes was present at a low level (25.9% of cases), low expression of peritumoral infiltrate PD-1 positive was observed in 44% of cases (68.7% in PDAC-associated UCOGC) [123]. Similar results were obtained later by Hrudka and colleagues. The authors also demonstrated that PD-L1 levels and tumor infiltrating lymphocytes are significantly associated to UCOGC as compared to conventional PDAC. Furthermore, PD-L1 levels define the patient’s prognosis. Indeed, UCOGC PD-L1 negative patients show a longer survival than UCOGC PD-L1 positive or PDAC (regardless of PD-L1 status) [124]. A recent paper also correlated PD-L1 expression with the *TP53* inactivation [123]. Nevertheless, this evidence was not confirmed by a later study, suggesting that this association is still controversial and should be better investigated [124]. In addition to PD-L1 expression, several monocytes/macrophages markers were analyzed in different works. A total of 80% of the UCOGC samples analyzed by Westra and collaborators displays positive staining for the monocyte CD68 (also called KP1) marker in specific mononuclear rich areas in UCOGC samples associated with *KRAS* codon 12 mutation [30]. Another group subsequently described the specific infiltration of mononuclear histiocytes (specific class 2 tumor-associated macrophages (TAM)) in UCOGC TME. Indeed, TAM2 population is the most represented class of TAM in pancreatic cancer; however, a specific upregulation of the TAM2 marker, CD163, is detected by IHC in all the analyzed UCOGC samples [123]. Moreover, another report also revealed the presence of the neutrophils infiltration in 40% of 15 cases analyzed through IHC [125].

In conclusion, as data concerning genetic/molecular characteristics of this variant are mainly obtained from small groups of patients or case reports, further studies are still necessary. Hence, given the rarity of the tumor, long term follow-up could improve not only the treatment of this variant, but also the literature necessary to define new strategies [126].

### 2.6. Rhabdoid Carcinoma

Another PDAC variant is the rhabdoid carcinoma. It is formally included in the group of UDC, but its important, both morphologic and genetic, peculiarities allow to present this tumor as a separate entity. This extremely rare variant is characterized by the presence of rhabdoid cells with peripheral nuclei, due to the presence of paranuclear filamentous structures and cytoplasmic eosinophilic inclusions (Figure 1) [37,127,128]. Agaimy and collaborators distinguish, among a cohort of patients which display a great percentage of rhabdoid cells (more than 50%), two subtypes: the subtype with pleomorphic cells and the subtype with monomorphic anaplastic cells. These two histological subtypes, in turn, correlate with specific genetic characteristics. The pleomorphic subtype displays *KRAS* alteration, due to allele mutation or copy number alteration, with 54% and 38% of patients, respectively, and an intact switch/sucrose non-fermentable (SWI/SNF) related, matrix associated, actin dependent regulator of chromatin, subfamily B (SMARCB) 1 by IHC. The monomorphic subtype shows a *KRAS*-wt and the loss of SMARCB1. However, taking advantage of a systematic review of the literature, the authors also highlight that the SMARCB1 analysis is missing in the 46 analyzed cases, with the exception of one patient who shows a *SMARCB1* missense mutation in the rhabdoid component [37]. SMARCB1 represents a core subunit of the SWI/SNF complex involved in chromatin remodeling. The one and only study showing *SMARCB1* mutation in a pancreatic mucinous carcinoma harboring rhabdoid features dates back to 2006 [129]. Tessier-Cloutier and collaborators recently analyzed the status of the expression of SWI/SNF protein in multiple UDC, including pancreatic UDC. However, despite the loss of SWI/SNF protein complex is mainly associated with rhabdoid features, the loss of SMARCB1 is observed only in 5% of cases, none of which were pancreatic cancer [130]. Equal results were previously obtained by Agaimy and collaborators. Indeed, the authors demonstrated that 92% of patients with rhabdoid cells, but not related to pancreatic cancer, displayed the loss of at least one component of the SWI/SNF complex [131]. Although no data are currently available to correlate *SMARCB1* mutation and rhabdoid pancreatic cancer, its role in other rhabdoid tumors is well established, thus suggesting that much effort should be made to uniquely define whether *SMARCB1* status also correlates with pancreatic rhabdoid or is a crucial difference between pancreatic cancer and other rhabdoid tumors [129,132].

### 2.7. Hepatoid Carcinoma

The hepatoid variant represents an extremely rare pancreatic adenocarcinoma that looks like hepatocellular carcinoma (Figure 1). An abundant eosinophilic cytoplasm characterizing cords of polygonal cells could be used for the diagnosis, whereas serum levels or tissue staining for the conventional hepatocarcinoma marker alpha-fetoprotein (AFP) still have a controversial role [38,39]. An interesting recent work by Xue and colleagues carried out on 163 resected pancreatic neuroendocrine tumors establishes the hepatoid variant as one of the most aggressive group after a classification considering type, frequency, and clinicopathological grade [133].

Given the rarity of this variant, in order to better know the morphological, histological, and pathological features, scientists can only take into account some case reports and literature reviews of the known cases. For example, a 2003 systematic literature review reports the main properties of the hepatoid variant, such as the localization principally in the head and the body of the pancreas, and the size greater than 5 cm at the time of diagnosis [38]. The first description of hepatoid variant molecular characteristics can be attributed to Vanoli and collaborators. The authors describe a case of hepatoid pancreatic tumor in a patient with elevated AFP serum levels. In addition to the canonical markers of hepatoid carcinoma (i.e., Hepar-1, AFP, and glypican-3), IHC results show an elevated reactivity with p62 in cytoplasmic eosinophilic globules [134]. Contrary to well-known markers, the main hallmark of hepatocellular carcinoma differentiation arginase I is not sufficient to discriminate between hepatocellular carcinoma and hepatoid adenocarcinoma. Indeed, Chandan and coworker analyzed the level of arginase I in several adenocarcinomas with multiple histological origins, including the pancreas. IHC analysis detected the presence of arginase I in 62.5% of cases. More in particular, the pancreatic hepatoid adenocarcinoma displays a focally positive distribution of arginase I [135].

Despite NGS molecular profiling is an increasingly feasible technique for both costs and simplicity of analysis, the rarity of this variant makes it almost impossible to genetically define it. The first case report that describes the specific mutation in hepatoid pancreatic adenocarcinoma dates back to 2016. More in particular, the patient showed two specific mutations in *BRCA1 associated protein* (*BAP*)*1* and *Notch1*. *BAP1* displayed a Q590fs frameshift mutation in exon 14. This gene encodes for a tumor suppressor involved in several processes such as DNA damage repair and transcriptional regulation, and is usually associated with a poor prognosis in different tumor types [136]. Notch1, instead, shows a rare single nucleotide polymorphism (i.e., A1343V) and encodes for a protein involved in cell to cell interaction [137]. Notch1 is one of the Notch transmembrane receptors; despite several studies carried out on mice describe both its tumor-suppressor and oncogenic function in pancreatic cancer, the activation of Notch1 is not sufficient to induce tumorigenesis [138]. However, the level of Notch1 is increased in PDAC tissue and a recent study performed on 50 patients with metastatic PDAC showed *Notch1* mutations in the 8% of cases [139,140].

### 2.8. Signet-Ring Cell Carcinoma

SRCC of the pancreas resembles the more common gastric counterpart and is composed of at least 80% of histologically-proven poorly-cohesive cells with well-defined cytomorphological characteristics. Indeed, scattered non-cohesive cells with vacuolated cytoplasm and irregular nuclei are detected in this histotype; the signet ring appearance of these cells is due to the cytoplasmic mucin that moves the nuclei toward cell periphery (Figure 1) [141]. Of all cases of pancreatic cancer, SRCC occurs with the lowest frequency (<1%): consistently only few cases are reported in the literature [40,142,143,144,145,146,147,148]. Signet-ring cells lack cell-to-cell interactions with high ability of invasion and stroma-infiltration, hence rendering the prognosis of this carcinoma among the worse. Despite all patients with pancreatic SRCC displaying a very rapid progression of the disease, Nepuri and colleagues reported the case of one patient displaying a good response to neoadjuvant therapy and reduced tumor size [40]. In 2018, Patel and colleagues published the first population-base analysis of primary pancreatic SRCC, in which the authors analyzed around 500 patients’ outcomes collected for 40 years. This study confirmes that SRCC displays poor prognosis and shows that age, primary site, and stage are significantly associated with both OS and disease specific survival (DSS), highlighting the urgent need for new biomarkers with potential clinical impacts for the treatment of this variant [41]. Indeed, mainly due to small sample size, no data about the genetic alterations in this variant are currently available. A personalized medicine strategy for these patients is, hence, not yet defined. Nevertheless, some molecular mechanisms underlying the formation of SRCC have been partially highlighted in different histotypes (e.g., stomach, prostate, and breast cancer) [149]. For instance, Fukui summarizes the current knowledge about the status of the signaling cascades involved in SRCC. In particular, phosphoinositide 3-kinase (PI3K) is constitutively activated in the signet-ring cells, due to the heterodimerization of ERBB2/ERBB3 complex. The activation of the kinases mitogen-activated protein kinase kinase (MEK) 1 and p38 disrupts the adherent junctions between the cells. Moreover, PI3K also affects the increased production of mucins (cause of the signet-ring appearance of this variant). Interestingly, the loss of cell-to-cell contact leads to the interaction between ERBB2 in the basolateral membrane and MUC4 in the apical membrane of the cells [150]. Indeed, only in this context the two molecules can physically interact with each other in order to further increase the phosphorylation and activation of ERBB2/ERBB3 complex [149].

## 3. Prospective for Targeted Therapy

The epithelial PDAC is one of the most lethal malignancies, mainly due to the high resistance to current therapies. Indeed, as opposite to the other solid cancers, the mainstay of PDAC treatment is still a conventional chemotherapy, such as the gemcitabine plus nab-paclitaxel combination regimen [151]. Nonetheless, these drugs display significant toxicities hence highlighting the urgent need to define precision medicine and improve the patient’s OS [152]. Furthermore, as we described before, not all PDAC are similar according to not only histopathological features but also genetic/molecular landscape [100,153]. This classification tragically complicates the design of new clinical trials, given the diversity between each rare histotype, and patients with these variants are not included in new clinical trials. A deeper characterization of these subtypes is hence fundamental to improve therapeutic choices in the clinical practice [154]. Furthermore, through the precious development of new generation of high-throughput sequencing techniques, it is possible to not only improve the knowledge of the landscape of molecular alterations, but even classify these tumor histotypes according to non-histological characteristics. Interestingly, recent evidence highlights that, similar to other cancer histotypes, transcriptomic subtyping is paving the way for a new era of PDAC precision medicine. Indeed, by non-negative matrix factorization (NMF) and separating tumor, stromal, and normal genes, Moffit et al. were able to actually integrate information of the heterogeneous pancreatic cancer cells with their intermixed abundant stroma. The authors distinguished two tumor-specific subtypes and two stromal-specific subtypes, and they correlated these subtypes with patient clinical outcome. Indeed, the tumor-specific basal-like tumors and the classic-subtypes display opposite clinical outcomes: the basal-like subtype more benefits from adjuvant therapy compared to the classical-subtype. Similarly, the stroma-specific activated-subtype has a worse median OS than the other stroma-specific normal-subtype [155,156]. Furthermore, through a compelling analysis of different papers reported in the literature, Collison and collaborators drew an outstanding phylotranscriptomic tree for pancreatic cancer subtypes [42]. Waddell and collaborators classified PDAC samples also according to structural rearrangements of the genome, as those alterations represent mechanisms of genomic damage. In particular, PDAC genomes are divided in four subtypes: (I) stable (20%); (II) locally rearranged (30%); (III) scattered (36%); and (IV) unstable (14%). This classification crucially confirms that patients with unstable genomes, which are hence susceptible to DDR, could actually benefit from drugs, such as platinum and poly (ADP-ribose) polymerase (PARP) inhibitors [157].

Among all the genetic alterations in PDAC, *BRCA* is undoubtedly the main actor which revealed the most important therapeutic implications in PDAC precision oncology, as shown by the wide use of the PARP inhibitors in cancer carrying *BRCA1/2* mutations. Indeed, the efficacy of olaparib in tumors with different histological origins (e.g., prostate, breast, and pancreatic cancer) and *BRCA* germline mutations have been reported since 2015 [158]. The mechanism by which PARP inhibitors act is the induction of synthetic lethality, as demonstrated in in vitro malignant pleural mesothelioma [159]. Very interestingly, a recent work by Golan and her group demonstrated that metastatic pancreatic cancer patients with *BRCA* germline mutation had longer progression free survival (PFS) when they received first-line platinum-based chemotherapy followed by olaparib treatment [160]. Unfortunately, a recent case report showed that despite an initial response to olaparib, new germline mutation of *BRCA2* occurr during the pressure of PARP inhibitor treatment. This acquired mutation leads to the restoration of DNA damage repair, thus leading to olaparib-resistance [161].

Mutations in *BAP1*, as well as those in *BRCA2*, lead to a DNA homologous recombination deficiency. Therefore, it is not surprising that chemotherapy causing genome instability is the best treatment to counteract cancer progression even in the presence of *BAP1* alterations. Consistently, a recent paper showed that OS is significantly high in patients with pleural mesothelioma carrying *BAP1* germline loss-of-function mutation, treated with platinum chemotherapy [162]. Similar to the PARP inhibitors, other molecules inhibiting the DNA polymerase subunits were identified. Although some inhibitors display their biochemical properties on polymerase a, b and e, in 2003 a paper on sulphoquinovosyl diacylglycerol first displayed its specific activity on POLE [163]. Nevertheless, no preclinical evidence is currently available for this class of agents.

Similar to the BRCAness tumors, other DNA-repair defects such as *ATM* inactivation significantly improve the sensitivity to chemotherapy and PARP inhibitors, through the synthetic lethality mechanism [164,165]. Interestingly, several compounds inhibiting ATM were generated during the time. In that respect, the two selective ATM inhibitors, AZD0156 and AZD1390, increase cell cycle arrest and apoptosis, as demonstrated by preclinical and early clinical phase evidence [166,167]. Notably, MMR and MSI are other cornerstones of genomic stability and therefore of new therapeutic strategies. Intriguingly, these two genomic alterations display a promising response to immunotherapies, in particular to the immune checkpoint inhibitors [168]. In that respect, for the first time, in 2017, the Food and Drug Administration approved the use of the PD-1 inhibitor pembrolizumab for the treatment of MSI or MMR deficiency solid tumors in adult and pediatric patients [169].

In addition to these conventional treatments, fully investigated and validated, even the other altered signaling cascades described in Table 2 are, at least in part, potentially druggable. Despite the fact that for these molecules only in vitro data or sometimes in other tumor histotypes are available, in this section we will mention the most important data, in order to provide the reader a general view of their potential implications in the clinical setting.

For example, many inhibitors of the protein kinase PKA or the second messenger cAMP, belonging to GNAS pathway, were discovered [56]. Nevertheless, despite the hyperactivation of this molecular pathway is well established in both pancreatic and other tumors, only in vitro data are currently reported for these agents [170]. Indeed, a very nice paper reported that the treatment with the activator of adenylyl cyclase enzymes forskolin hinder PDAC cell motility, by enhancing cAMP levels and F-actin remodeling [53]. The PKA-downstream YAP and WW-domain-containing transcription regulator 1 (TAZ) also display oncogenic features in PDAC, through switching off the Hippo tumor suppressor pathway [171]. Consistently, Rozengurt and his colleagues demonstrated that the activation of pathways inhibiting YAP (e.g., liver kinase B1 (LKB1) and mammalian target of rapamycin complex (mTORC)1) favors a better prognosis, hence suggesting the important implications of YAP targeting in PDAC clinical outcome [172].

Given the high ability of pancreatic cancer cells to invade healthy tissues and metastasize, several in vitro studies investigated the role of multiple molecules in regulating the re-expression of the cell adhesion molecule E-cadherin and/or the well-known EMT process. For example, Yuan and collaborators showed that α-mangostin, a xantone compound, induces the expression of E-cadherin in pancreatic cancer cell lines [173]. Similarly, anthothecol, a limonoid used for antimalarial treatment, suppresses both pancreatic cells motility and invasion through the induction of E-cadherin expression, and N-cadherin and ZEB1 inhibition, when it is given as anthothecol-encapsulated poly lactic-co-glycolic acid (PLGA)-nanoparticles [174]. E-cadherin expression can be also modulated by the regulation of epigenetic machinery, as demonstrated by Von Burstin and his colleagues. Indeed, the authors showed that trichostatin A, an inhibitor of histone deacetylase (HDAC), acts on *CDH1* gene, in turn restoring E-cadherin expression in in vitro systems [175]. Moreover, curcumin reverses the EMT process by not only downregulating E-cadherin expression, but also inhibiting the HH pathway [176]. Multiple HHs’ target inhibitors are nowadays developed and despite Smo is one of the main druggable-target, several inhibitors for GLI are currently under investigation. Nevertheless, no GLI3-specific inhibitor is actually validated [177].

Regarding the membrane receptor Notch1, a recent paper demonstrated that Notch1 loss of function increases the sensitivity to PI3K/mTOR pathway inhibitors in in vitro head and neck squamous cell carcinoma [178]. Moreover, although not always specific for Notch1, antibodies developed to antagonize dysregulated Notch represent a promising approach of immunotherapy in several solid cancers [179]. Nevertheless, Notch1 inhibition mainly occurs through the use of γ-secretase inhibitors [180]. Interestingly, the use of two γ-secretase inhibitors, RO4929097 or DAPT, impair tumor growth in prostate cancer cell lines [181]. Moreover, DAPT synergistically acts with gemcitabine, by counteracting gemcitabine-induced stemness in pancreatic cancer cells. This evidence highlights the potential role of γ-secretase inhibitors in overcoming chemoresistance of pancreatic cancer patients [182]. Indeed, Notch1 status represents a negative predictive biomarker of gemcitabine treatment and prognosis in pancreatic cancer patients [183].

## 4. Conclusions

In recent years, precision oncology has undoubtedly been one of the most important revolutions in clinical practice and has significantly improved the quality and life expectancy of cancer patients. Despite the attempt to untie the Ariadne’s string connecting genetic/molecular landscape of pancreatic cancer cells with their complex TME, PDAC and its variants remain the deadliest tumors in the Western world [5]. Furthermore, the classification of PDAC variants according histopathological characteristics still do not help clinicians in the best treatment choice for personalized medicine. Defining non-histopathological features, such as genetic and transcriptomic information, is currently under investigation and could actually be the keystone in building new rationale, in order to obtain the best pharmacological responses and PDAC patients’ OS.

## Figures and Tables

**Figure 1 ijms-21-08841-f001:**
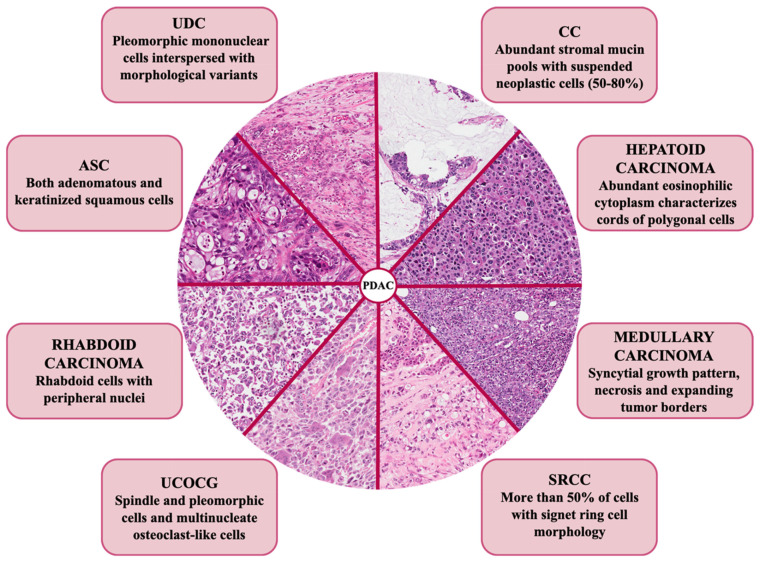
Histologic hallmarks of the different morphologic pancreatic ductal adenocarcinoma (PDAC) variants. Colloid carcinoma (CC): this variant shows extracellular pools of mucin with floating neoplastic cells. Hepatoid carcinoma: this variant is very similar to hepatocellular carcinoma, with polygonal cells showing eosinophilic and large cytoplasm. Medullary carcinoma: this variant shows a syncytial-growth pattern, without glandular structures. Signet-ring cells carcinoma (SRCC): in this variant, neoplastic cells are poorly cohesive and show intracellular mucin with peripheral nuclei. Undifferentiated carcinoma with osteoclast-like giant cells (UCOGC): this variant is composed of three types of cells: multi-nucleated osteoclast-like cells, which are giant cells belonging to the monocyte-macrophage lineage, mononuclear histiocytes and neoplastic cells, which are usually very atypical. Rhabdoid carcinoma: the cells are very atypical and enlarged, with classic rhabdoid features. Adenosquamous carcinoma (ASC): in this variant, there is a mixture of neoplastic glands and tumor cells with squamous features (to be classified adenosquamous, the adenocarcinoma must contain at least 30% of squamous component), with the latter that may have a high degree of atypia. Undifferentiated carcinoma (UDC), anaplastic subtype: this variant totally lacks a glandular architecture and tumor cells are very atypical and pleomorphic; UDC, sarcomatoid variant: this variant totally lacks a glandular architecture and neoplastic cells are very atypical and spindle-shaped. Original magnification: ×20.

**Table 1 ijms-21-08841-t001:** Genetic status in canonical driver genes, overall incidence, and prognosis of each pancreatic cancer variants.

Subtype	*KRAS*	*TP53*	*SMAD4*	*CDKN2A*	Incidence	Prognosis	Ref
CC	30–50%	20%	n.a.	n.a.	1–3%	Good (5-year survival >55%)	[9,10,11,12]
Medullary carcinoma	17–30%	18%	n.a.	n.a.	<1%	Still unclear	[13,14,15,16]
ASC	90–100%	50–90%	18->90%	6%	1–4%	Poor (9 months from the diagnosis)	[17,18,19,20,21,22,23]
UDC	30–70%	30–60%	n.a.	n.a.	<1%	Extremely poor (5 months from the diagnosis	[24,25,26,27,28,29]
UCOGC	70–100%	50–100%	10–50%	25%	<1%	Better than UDC	[24,25,26,27,30,31,32,33,34,35,36]
Rhabdoid carcinoma	40%	40–50%	n.a.	n.a.	<1%	Still unclear	[37]
Hepatoid carcinoma	n.a.	n.a.	n.a.	n.a.	<1%	Still unclear	[38,39]
SRCC	n.a.	n.a.	n.a.	n.a.	<1%	Poor	[40,41]

ASC, adenosquamous carcinoma; CC, colloid carcinoma; n.a., not assessed; SRCC, signet-ring cell carcinoma; UCOGC, undifferentiated carcinoma with osteoclast-like giant cells; UDC, undifferentiated carcinoma.

**Table 2 ijms-21-08841-t002:** Non-canonical genetic and molecular alterations in pancreatic cancer variants.

Subtype	Gene	Chromosome	Mutations	Encoded Protein	Functional Effects on Molecular Pathway
CC	*GNAS*	20	Somatic	Gα subunit of heterotrimeric G-proteins	GPCR-mediated signaling constitutively active
	*ATM*	11	Germline	Serine/threonine kinase	DNA double strand break not tagged
	*MLH1*, *MLH2*, *PMS2, MSH6*	3, 2, 7, 2	Germline	Protein-protein interactions in MMR	MSI
Medullary carcinoma	*MLH1*, *MLH2*, *PMS2, MSH6*	3, 2, 7, 2	Germline	Protein-protein interactions in MMR	MSI
	*POLE*	12	Somatic	Catalytic subunit of the DNA polymerase	DNA damage
ASC	*UPF1*	19	Somatic	RNA helicase	Altered NMD
	*KMT2C, KMT2D, SMARCA4, KDM6, KDM3*	7, 12, 19, X, 2	n.a.	Chromatin modifiers	Altered chromatin architecture
UDC	*CDH1*	16	n.a.	E-cadherin, cell adhesion molecule	EMT
UCOGC	*SERPINA3*	14	Somatic	α−1antichymotrypsin	n.a.
	*MAGEB4*	X	Somatic	Cancer antigen	n.a.
	*GLI3*	7	Somatic	Transcription factor	Constitutive activation of HH signaling
	*MEGF8*	19	Somatic	n.a.	Constitutive activation of HH signaling
	*TTN*	2	Somatic	Muscle assembly and functioning	n.a.
	*BRCA2*	13	Somatic	Rad51 binding protein	DNA damage
Rhabdoid carcinoma	*SMARCB1*	22	Somatic	INI1	Chromatin remodeling (BAF complex)
Hepatoid carcinoma	*BAP1*	3	Somatic or germline	Ubiquitin carboxyl-terminal hydrolase	DNA damage
	*Notch1*	9	n.a.	Membrane receptor	Alteration in cell to cell interactions
SRCC	n.a.	n.a.	n.a.	n.a.	PI3K and MEK1 constitutively active

ASC, adenosquamous carcinoma; ATM, ataxia telangiectaisa mutated; BAP1, breast cancer gene 1 (BRCA1) associated protein 1; BRCA2, breast cancer 2; CC, colloid carcinoma; GPCR, G protein-coupled receptor; EMT, epithelial-to-mesenchymal transition; GLI3, glioma-associated oncogene 3; HH, hedgehog; KDM, lysine demethylase; KMT2, histone-lysine N-methyltransferase 2; MAGEB4, melanoma-associated antigen B4; MEGF8, multiple epidermal growth factor-like domains protein 8; MEK1, mitogen-activated protein kinase kinase 1; MMR, mismatch repair; MSI, microsatellite instability; n.a., not assessed; NMD, nonsense-mediated mRNA decay; PI3K, phosphoinositide 3-kinase; POLE, polymerase epsilon; SERPINA3, serpin peptidase inhibitor clade A member 3; SMARCB1, switch/sucrose non-fermentable (SWI/SNF) related, matrix associated, actin dependent regulator of chromatin, subfamily B, 1; SRCC, signet-ring cell carcinoma; UCOGC, undifferentiated carcinoma with osteoclast-like giant cells; UDC, undifferentiated carcinoma; UPF1, up-frameshift 1.

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
