# Peer review of "Morphologic and Molecular Landscape of Pancreatic Cancer Variants as the Basis of New Therapeutic Strategies for Precision Oncology"

_ijms, 2020, doi:10.3390/ijms21228841_

Round 1
Reviewer 1 Report
Bazzichetto and group have summarized the new landscape of pancreatic cancer subtypes and their molecular and morphological features. They also included the mutation status and basis for new therapeutic agents. It is very interesting article and can be accepted after revision. Please add following information to the review
- What is the KRAS, p53, TGFbeta receptor mutation status in different subtypes?
- What is the survival status in PDAC patients with different subtypes?
- What is the preference for metastasis for PDAC these subtypes?
- In tumor microenvironment of these different subtypes, is there difference in fibroblasts or immune cells?
- If possible can you include the table of how many percent cases we see for each subclass and which is most deadly?
Author Response
Dear Editor,
First, we would like to thank the Reviewers for their helpful and constructive comments, and their recognition of the potential relevance of our review; we have tried our best to respond to all of them and believe that the revised manuscript has now substantially improved. We thus hope it can be considered for publication in the Special Issue entitled “Pancreatic Ductal Adenocarcinoma: Precursors and Variants” in International Journal of Molecular Sciences.
As suggested by the Reviewers, we modified the manuscript as following:
Reviewer #1:
- What is the KRAS, p53, TGFbeta receptor mutation status in different subtypes?
We thank the Reviewer for this point. We added a new table (new Table 1), in which all this information and relative references are reported. We want to highlight that some of these variants occurs at a very low incidence, hence available data are few, and the reported percentages often include only few patients. Furthermore, at the best of our knowledge, despite it would be a crucial point due to the relevance of TGF-b/TGF-b-R axis in PDAC tumor microenvironment, we didn’t find any reports about TGF-b receptor status in none of these variants.
- What is the survival status in PDAC patients with different subtypes?
Due to the very rarity of these variants, data on survival are few. Only for some of these variants, specific information about survival status is more defined. We added available data and related references in the new Table 1.
- What is the preference for metastasis for PDAC these subtypes?
Pancreatic cancer commonly metastasizes in liver and lymph nodes. Similar to PDAC, none of the described variants display metastases in specific tissues and organs. Interestingly, a recent manuscript reports a man with pancreatic UCOGC, with metastatic lesions in the lung. As the poorly differentiated adenocarcinoma cells expressed PD-L, clinicians gave pembrolizumab therapy. Albeit pembrolizumab regimen achieved limited effects on primary pancreatic cancer, it displayed antitumor properties against lung metastases (Obayashi M. et al., 2020, 10.1186/s12876-020-01362-4).
- In tumor microenvironment of these different subtypes, is there difference in fibroblasts or immune cells?
We thank the Reviewer for discuss this important point, as pancreatic cancer is among the most fibrotic cancer characterized by a complex tumor microenvironment. As suggested, we added the TME characteristics of the two medullary and UCOGC variants. Indeed, at the best of our knowledge, only medullary carcinoma and UCOGC display specific features in terms of the number of lymphocyte and TAM infiltration, respectively.
- If possible can you include the table of how many percent cases we see for each subclass and which is most deadly?
We added the requested information in the new Table 1, which now reports also the percentage of incidence of each variants and relative references. UDC and ASC are the most deadly PDAC variants, as previously reported for the prognosis of the variants.
Reviewer 2 Report
Bazzichetto et al. comprehensively reviewed molecular/genetic and histopathological characteristics of pancreatic ductal adenocarcinoma (PDAC) variants, recently classified by WHO. Although more studies are required to link between genetics and these variants, this is a timely review to provide an overview on our current understanding of PDAC variants. The manuscript has been nicely written, and issues are well summarized. Minor points and suggestions are provided as below.
Minor points
- The authors nicely presented histopathology of each variant in Figure1. It would be better to provide a representative H&E image for each variant in each section with the detailed information. This would be helpful for readers to appreciate its molecular characteristics and histopathology.
- There are recent genetic alterations associated with adenosquamous subtype of PDAC such as KDM3A, FGFR-1ERLIN fusion, KMT2C, SMARCA4, KDM6A [1,2]. It is worth including these recent findings.
- While this review largely focuses on histophatological variants, it is worth mentioning in the discussion section about other molecular subtypes based on transcriptomic analysis. In the future, it is necessary that the classification of histopathology and molecular subtypes based on transcriptome need to be comprehensively integrated to better understand this disease. Although we don’t have a conclusive answer for this in current knowledge, the authros might need to discuss this aspect in their discussion session.
- Greek words are missing or not properly used in the manuscript.
- In line 307, ‘we have to wait until 2010’ need to be past tense.
- In line 391, ‘should be better investigate’ à ‘should be better investigated’
1 Lenkiewicz, E. et al. (2020) Genomic and Epigenomic Landscaping Defines New Therapeutic Targets for Adenosquamous Carcinoma of the Pancreas. Cancer Res DOI: 10.1158/0008-5472.can-20-0078
2 Hayashi, A. et al. (2020) A unifying paradigm for transcriptional heterogeneity and squamous features in pancreatic ductal adenocarcinoma. Nat Cancer 1, 59–74
Author Response
Dear Editor,
First, we would like to thank the Reviewers for their helpful and constructive comments, and their recognition of the potential relevance of our review; we have tried our best to respond to all of them and believe that the revised manuscript has now substantially improved. We thus hope it can be considered for publication in the Special Issue entitled “Pancreatic Ductal Adenocarcinoma: Precursors and Variants” in International Journal of Molecular Sciences.
As suggested by the Reviewers, we modified the manuscript as following:
Reviewer #2:
- The authors nicely presented histopathology of each variant in Figure1. It would be better to provide a representative H&E image for each variant in each section with the detailed information. This would be helpful for readers to appreciate its molecular characteristics and histopathology.
First, we thank the Reviewer for his/her recognition of the Figure 1. Our goal is to give to the reader an overview of the difference in histopathological features of the PDAC variants. Hence, we would like to maintain this comprehensive figure. However, as kindly suggested by the Reviewer, in order to help the reader in appreciating the histopathology, we improved the specific description of each variant. Similarly, we added new references.
- There are recent genetic alterations associated with adenosquamous subtype of PDAC such as KDM3A, FGFR-1ERLIN fusion, KMT2C, SMARCA4, KDM6A [1,2]. It is worth including these recent findings.
We appreciate the Reviewer’s comment, and we thank him/her for suggesting us these important molecular characteristics reported in adenosquamous variant. The most relevant data reported in these two recent papers are now added in the main text.
- While this review largely focuses on histophatological variants, it is worth mentioning in the discussion section about other molecular subtypes based on transcriptomic analysis. In the future, it is necessary that the classification of histopathology and molecular subtypes based on transcriptome need to be comprehensively integrated to better understand this disease. Although we don’t have a conclusive answer for this in current knowledge, the authors might need to discuss this aspect in their discussion session.
We thank the Reviewer for highlighting this truly important aspect, which otherwise would have remained undisputed in this manuscript. Indeed, recent published reports clearly emphasize the great potential of the new highthroughput techniques in classifying PDAC by transcriptomic features, even by distinguishing cancer cells and the surrounding stromal cells (particularly abundant in fibrotic pancreatic cancer). The main evidence and relative references are now added in the main text in the “Prospective for targeted therapy” section.
- Greek words are missing or not properly used in the manuscript.
- In line 307, ‘we have to wait until 2010’ need to be past tense.
- In line 391, ‘should be better investigate’ à ‘should be better investigated’
We thank the Reviewer for suggesting us some typos mistakes. All the grammatical inaccuracies are now corrected through the text and we added new references
On behalf of all authors,
Yours sincerely,
Fabiana Conciatori
Round 2
Reviewer 1 Report
All of my questions are answered. I recommend acceptance.